# Effects of Greenshell Mussel (*Perna canaliculus*) Intake on Pathological Markers of Multiple Phenotypes of Osteoarthritis in Rats

**Parkpoom Siriarchavatana [1,\*]** , **Marlena C. Kruger [2,3]** , **Matthew R. Miller [4]** ,
**Hong (Sabrina) Tian [5]** and **Frances M. Wolber [1,6]**

1   School of Food and Advanced Technology, Massey University, Palmerston North 4442, New Zealand;
    f.m.wolber@massey.ac.nz
2   School of Health Sciences, Massey University, Palmerston North 4442, New Zealand;
    m.c.kruger@massey.ac.nz
3   Riddet Centre of Research Excellence, Massey University, Palmerston North 4442, New Zealand
4   Cawthron Institute, Nelson 7042, New Zealand; matt.miller@cawthron.org.nz
5   Sanford Ltd., Auckland 1010, New Zealand; STian@sanford.co.nz
6   Centre for Metabolic Health Research, Massey University, Palmerston North 4442, New Zealand
\*   Correspondence: P.Siriarchavatana@massey.ac.nz; Tel.: +64-21-142-6234

**Abstract:** The prevalence of metabolic osteoarthritis has been increasing worldwide, particularly among women. The aim of this study was to investigate the effects of the New Zealand greenshell mussel (*Perna canaliculus*; GSM) on osteoarthritis (OA) prevention in a rat model. One-hundred-and-eight female rats aged 12 weeks were divided into four test groups, containing 24 rats each, plus an additional control group. Each test group received one of the four experimental diets: normal control diet (ND), normal control diet supplemented with GSM (ND + GSM), high fat/high sugar diet (HFHS), or high fat/high sugar diet supplemented GSM (HFHS + GSM), for 36 weeks (end of the study). After 8 weeks on experimental diets, half of each group was subjected to ovariectomy (OVX) and the remaining half received a sham operation (ovaries left intact). The study evaluated body composition, bone mass, plasma cytokines, adipokines, HbA1c, CTX-II, and knee joint's histopathology. HFHS diet and OVX significantly induced body weight gain and leptin production. OVX rats lost bone mineral density but increased adiponectin, HbA1C, and MCP-1. The OVX rats fed HFHS showed the highest Mankin scores. Importantly, inclusion of GSM reduced these pathological features. In conclusion, GSM might be beneficial in halting the progression of OA.

**Keywords:** metabolic osteoarthritis; obesity; menopause; animal models; greenshell mussel

## 1. Introduction

Osteoarthritis affects people worldwide, especially those in advanced age. Currently, the global prevalence of knee OA is 3.7% of the population, which means 268 million people are suffering from OA in this particular joint [1]. However, OA can occur in any joints including hip, fingers, hands, wrist, and temporomandibular joints [2]; thus, the total prevalence of OA is greater than the previous estimation. The United Nations and World Health Organization placed this burden as the fourth leading cause of disability in 2020 [3].

The disease is characterized by articular cartilage deterioration, which progresses slowly and appears initially at a subclinical level, making it difficult to be detected at an early stage [4]. Therefore, patients diagnosed with OA already have some severity of cartilage erosion. Treatment is also challenged by the fact that avascular tissue such as cartilage has less ability to regenerate itself [5].

Moreover, microstructural damage to subchondral bone is replaced by fibrous tissue resulting in less porosity of subchondral bone plate, which limits the nutrient supply to the cartilage layers [6,7]. The ideal strategies to alleviate the disease burden would focus on both prevention and halting disease progression rather than relieving pain by anti-inflammatory drugs. Effective treatments need to target the underlying causes of diseases but the etiology of OA is very complicated, especially in the case of metabolic osteoarthritis (MetOA). It is not clear whether OA is a single disease itself or a nebula of several diseases which have no specific markers to identify them.

There are many animal models to study OA, including both spontaneous OA and induction OA models. The spontaneous OA model, using animals which are genetically prone to develop OA in advancing age [8,9], theoretically mimics the pathological mechanisms for age-related OA in humans, whereas surgical induction OA models mimic post-traumatic OA [10]. The incidence of MetOA has been increasing due to the global expansion of obesity and related chronic diseases [11,12], increasing the importance of developing and optimizing a reliable animal model for this OA phenotype. The use of a high energy diet due to increased fat or sugar proportion successfully induces obesity in rat models; however, the effectiveness of this model to generate OA is controversial. Some studies have revealed that high-fat alone was inadequate to cause cartilage lesions in a rodent model unless groove surgery was incorporated [13,14]. However, other studies demonstrated that the pathological lesions of MetOA were successfully established in rats by feeding a high-fat diet for 16 or 28 weeks [15–17]. The success of these studies might be related to the gender as male rats are prone to osteoarthritis because of the influence of testosterone while female rats had become prone to the disease only when they lack estrogen [18]. In addition, the variation in success rates also depends on the type and quantity of fat [19]. It therefore remains inconclusive whether OA is inducible in female rats by a high energy diet or ovariectomy.

This study focused on the MetOA in a female population, which is both the most prevalent phenotype and the most complicated. There is no established set of biomarkers to distinguish the phenotype or subtypes of OA. We hypothesized that different causes could activate MetOA in unique pathways resulting in distinguishable pathological patterns. Therefore, the experiment was designed to find the differences in pathological markers from individual causes and as well as additive causes. In addition, greenshell mussel (*Perna canaliculus*, GSM), a rich source of omega-3 fatty acid [20,21], showed some evidence supporting its use for anti-inflammation and arthritis [22–24]; however, the previous animal studies have been conducted using GSM oil employed chemical or surgical induction methods which obviously represent a post-traumatic OA phenotype rather than MetOA [22,24], which differ in cause and pathogenesis [25–27]. Our study, on the other hand, hypothesized that another preparation of GSM, flash-dried powder which may be more economically accessible for people compared to expensive oil extracts and which may contain other unknown bioactive compounds, could prevent MetOA induction or slow MetOA progression using a model that more closely mimics the human condition.

The animal model was designed to simulate the broad range of MetOA severity in the postmenopausal women by feeding the rats a Western-style diet high in sugar and high in saturated fat, and by ovariectomizing some of the animals to induce menopause. The GSM intervention was designed to mimic a reasonable human consumption pattern of one meal of mussels per day providing approximately one-third of the person's protein intake the results of the study provide insight in how single versus multiple causes could affect the pathological pattern in the rat model and identify the potential of GSM in preventing MetOA.

## 2. Materials and Methods

### 2.1. Greenshell Mussel (GSM) Powder

Flash-dried powder from whole GSM meat was produced by Sanford Ltd. (ENZAQ facility, Blenheim, New Zealand) using standard manufacturing processes and assessed for proximate

composition in a commercial testing laboratory (Food Testing Laboratory of Cawthron Analytical Services, Nelson, New Zealand). The GSM powder contains 8.1% fat, 43% crude protein, 21.9% carbohydrate, 5.8% moisture, and 21.2% ash. The fat contains 29.4% omega-3 fatty acid and 3.5% omega-6 fatty acid.

Ten percent of GSM powder was included in the experimental diets: normal control diet (ND + GSM) and high fat/high sugar diet (HFHS + GSM). The diet formulae were predetermined in order to normalize the energy in the diet counterpart. All diets were stored at −20 °C until used.

## 2.2. Animal Study

The overall experimental plan is present in Figure 1. One-hundred-and-eight female Sprague Dawley (SD) rats aged 11 weeks reared on standard chow were obtained from the Small Animal Production Unit at Massey University (Palmerston North, New Zealand) and the study was conducted in the same animal facility as approved by the Massey University Animal Ethics Committee (MUAEC protocol/approval 16/112). The animal room environment was set at 22 ± 1 °C, with 45–55% humidity, and a 12/12 light–dark cycle throughout the study. The rats were singly housed in conventional cages with heat-treated aspen wood shavings as bedding. After a one-week acclimatization period, the rats were randomized into test groups and fed one of four diets (Specialty Feeds, Glen Forrest, Western Australia): (1) normal control diet (ND) containing 5% total fat (from soy oil), 5% sucrose, and 15% total protein (from casein); (2) normal control diet supplemented with GSM (ND + GSM) containing 5% total fat (84% from soy oil/16% from GSM), 5% sucrose, and 15% total protein (66% from casein/33% from GSM); (3) high-fat high-sugar diet (HFHS) containing 30% total fat (50% from soy oil/50% from lard), 30% sucrose, and 15% total protein (from casein); and (4) high-fat high-sugar diet supplemented with GSM (HFHS + GSM) containing 30% total fat (49% from soy oil/49% from lard/1% from GSM), 30% sucrose, and 15% total protein (66% from casein/33% from GSM). After 8 weeks on the experimental diets (age 20 weeks), ninety-six rats were subjected to surgery: half of them were ovariectomized and the rest were left with ovaries intact as sham control animals. The surgical procedure and anesthesia were performed as described previously [28]. These surgically manipulated rats were on their experimental diets through to the end of the study (age 48 weeks) before termination and biological samples were collected. Twelve rats from a previous study which did not undergo any surgery that were fed the normal control diet were included in the data set in order to compare knee joint scoring between rats of different age. These were sacrificed at 26 weeks of age and knee joints were collected for histopathological slide preparation. All rats were euthanized by exsanguination following anesthesia.

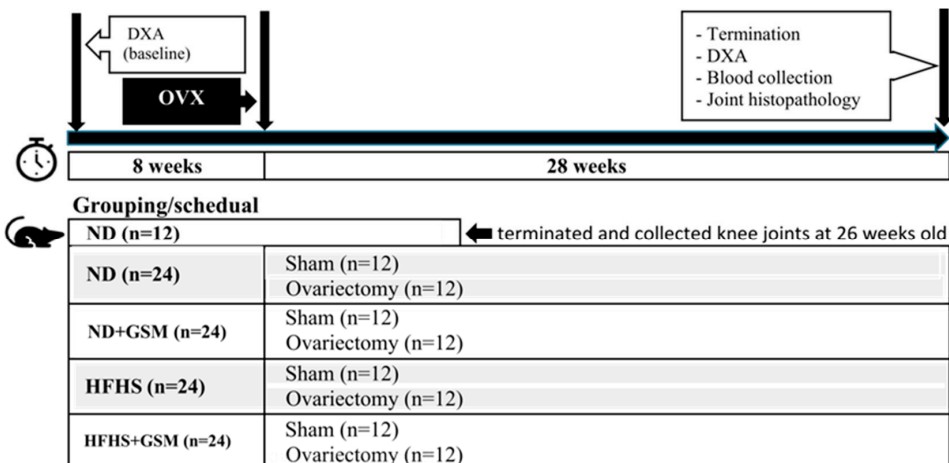

**Figure 1.** Experimental schedule for obesity-induced osteoarthritis in the rat model with greenshell mussel. DXA = Dual X-ray energy absorptiometry; OVX = ovariectomy.

### 2.3. Dual-Energy X-ray Absorptiometry (DXA) Scans

In vivo percent whole body fat and bone mineral content from lumbar spine and right femur were evaluated using a Hologic Discovery scanner (Hologic, Bedford, Massachusetts), at the beginning of the study and the end (age 48 weeks). Anesthesia was provided prior to DXA scans as described previously [28]. Daily quality control (QC) scans were run prior the scanning; the coefficient of variation (CV) for the spine was 0.98–1%, and the CV for the femur was 0.85–0.92%. Rats were positioned supine with right angles between the spine and femur. Three high resolution scans were applied for every bone site.

### 2.4. Plasma Analysis

At the end of the study, anesthetized rats underwent cardiac puncture. Blood was drawn and collected in EDTA-anticoagulant tubes. Plasma was separated by centrifugation at 2500× $g$ rpm (Herareus Megafuge 1.0R, Thermofisher) and kept at −80 °C until analysis. All analytes were measured by ELISA methods with different kits. IL1β, IL-6, IL-10, and TNF-α were assessed using Duo-set® kits (R&D system, Minneapolis, MI, USA). MCP-1, adiponectin, and leptin were assessed using Quantikine® kits (R&D). PGE2 was assessed using PGE2 ELISA kit (Invitrogen, Vienna, Austria). CTX-II assay kit was obtained from Cloud-Clone Corp, TX, USA. HbA1C assay kit was obtained from Fine Test, Wuhan Fine Biological Technology, Wuhan, China. Assays were performed following the manufacturers' instruction and optical density measured using a microplate reader (Multiskan FC, Thermo Fisher Scientific, Vantaa, Finland). The other inflammatory cytokines, INF-γ, CXCL-1, IL-12, IL-17, IL-18, and IL-33, were measured by LEGENDplexTM (BioLegend®, San Diego, CA, USA) using a Gallios Flow Cytometer (Beckman Coulter, CA, USA). The concentrations of the analytes were calculated against their standard curves accordingly to individual kits. All tests were performed in duplicate.

### 2.5. Leptin Gene Expression

Rats were dissected and perigonadal fat was harvested then immediately frozen in liquid nitrogen prior to storage in −80 °C until analysis. Tissue homogenization was performed as follows; 200 mg of fat tissue in 2 mL tubes with 1 mL of Tri reagent® was dissociated in a homogenizer (Precellys Evolution Homogenizer, Bertin). Two-hundred microliters of chloroform was added in the homogenate then shaken vigorously prior to centrifuging at 12,000× $g$ for 15 min at 4 °C. The clear upper part of the homogenate was recovered and further subjected to the column of the RNA extraction kit (Direct-zol™ RNA Miniprep Plus, Zymo Research, Irvine, CA, USA). cDNA was synthesized from the total RNA according to manufacturer's protocol using SuperScript™ IV First-Strand Synthesis System (Invitrogen, Carlsbad, CA, USA). Real-time quantitative PCR, using SYBR™ Green Master Mix, was performed on LightCycler® 480 Real-Time PCR instrument (Roche Applied Science). The specific primers for leptin were NM_013076.3 -CCAGGATCAATGACATTTCACA (forward); -AATGAAGTCCAAACCGGTGA (reverse). The PCR reaction was initialized with 1 cycle of 95 °C for 10 min, followed by 35 cycles of denaturation step at 95 °C for 20 s and extension step at 72 °C for 20 s. Expression levels were calculated relative to the mean of ND sham rats and GAPDH expression (Primers XM_017593963.1 forward-CTGCACCACCAACTGCTTAG; reverse-TGATGGCATGGACTGTGG) was used as the internal control. Quantitative measurements were determined using the (2-ΔΔCt) method. Amplification of specific transcripts was confirmed by melting curve analysis.

### 2.6. Histopathological Examination of Knee Joints

At the termination, knee joints were harvested and fixed in 10% buffer formalin followed by decalcification in 10% ethylenediaminetetraacetic acid and were later embedded in paraffin. Coronal section was applied from the medial collateral ligament and stained with Safranin-O. The pathological changes of the knee joints were assessed using the Mankin scoring method as described previously [29]. Both medial and lateral tibia were examined and the cumulative data of both were analyzed.

### 2.7. Statistical Analysis

Two-way ANOVA was used to evaluate the effects of diet or GSM or OVX on the following parameters: body composition, bone mineral content, plasma analytes, and leptin gene expression. One-way ANOVA was also applied to identify the differences across all groups and in case a significant difference was detected, a multiple comparison test, Tukey test, followed. Mann–Whitney U test was selected for knee joint pathological analysis. All analyses were performed using IBM statistic software version 25 (Armonk, NY, USA) and *p*-values < 0.05 were considered to be significant. All data are presented as mean ± SE.

## 3. Results

### 3.1. Changes in Body Composition

At the beginning of the study, the mean body weight of all rat groups ranged from 267.37 to 281.62 g with no significant differences (Table A1). Increased body weight of all rats was observed at the end of the 36-week study period and, as expected, HFHS alone significantly increased both body weight and body fat gain. Likewise, ovariectomy alone resulted in significantly higher weight gain and even greater changes in percent body fat. The success of ovariectomy was confirmed by recession of uterus at necropsy process (data not shown). The combination of ovariectomy and HFHS diet had additive effects and resulted in the greatest increases in both body weight and body fat.

### 3.2. Bone Mineral Density (BMD) of Lumbar Spine and Femur

BMD from anesthetized rats was measured using DXA at the end of the study (Figure 2). Lumbar spine displayed lower BMD than both sides of femur in all rats. In general, OVX rats had lower BMD than sham rats in all bone sites. Lumbar spine BMD in all sham rats was ~0.22 g/cm$^2$, while OVX rats displayed only 0.20 g/cm$^2$. Similarly, both left and right femur BMD in sham rats was ≥ 0.28 g/cm$^2$, while OVX rats showed ≤ 0.27 g/cm$^2$ and ≤ 0.26 g/cm$^2$ in the left and right femur, respectively. The significant differences were only noticed due the effect of OVX on all bone sites. No significant differences were detected comparing the cluster of ND with HFHS fed rats or the cluster of no GSM with GSM fed rats.

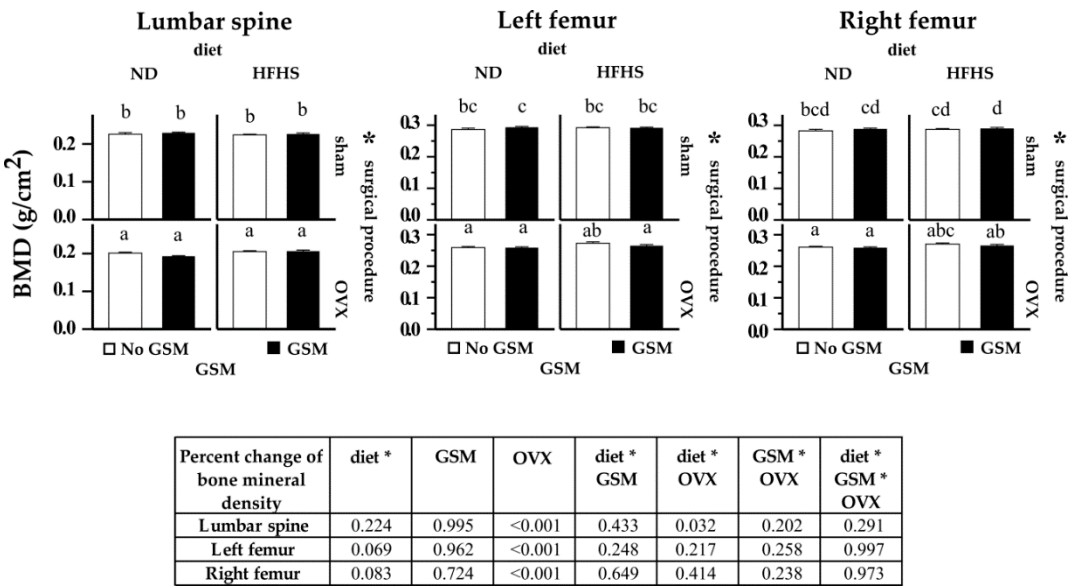

| Percent change of bone mineral density | diet * | GSM | OVX | diet * GSM | diet * OVX | GSM * OVX | diet * GSM * OVX |
|---|---|---|---|---|---|---|---|
| Lumbar spine | 0.224 | 0.995 | <0.001 | 0.433 | 0.032 | 0.202 | 0.291 |
| Left femur | 0.069 | 0.962 | <0.001 | 0.248 | 0.217 | 0.258 | 0.997 |
| Right femur | 0.083 | 0.724 | <0.001 | 0.649 | 0.414 | 0.238 | 0.973 |

**Figure 2.** Bone mineral density in lumber spine and femur (each group *n* = 11–12). Bone mineral density (BMD) was evaluated using dual energy X-ray absorptiometry at the end of the study (age 48 weeks). Bar graphs and error bars show means and SE respectively. Two-way ANOVA was used to evaluate the influence of effects; diet, surgical procedure, and GSM. If the effect was statistically significant, the asterisk * is indicated. One-way ANOVA was used to identify significant differences across all groups. In case data reached the criteria, a multiple comparison test is performed using Turkey method, and then letters are indicated on the bar. Groups showing different letters are significant differences. Table in the figure shows *p* values calculated from two-way ANOVA analysis. *p* values less than 0.05 was used to determine statistical significance in all methods.

### 3.3. Analysis of Inflammatory Markers in Systemic Circulation

All inflammatory markers are presented in Table A2. Most proinflammatory and anti-inflammatory cytokines in plasma were present at very low concentrations and not influenced independently by diet, GSM, or surgery. MCP-1, however, was significantly increased in the OVX rats when compared to the sham rats. Moreover, the increased MCP-1 levels in the ovariectomized rats fed HFHS were different from ND although a significant diet effect was not detected. There was also an interaction effect of GSM and OVX on IL-17. The result showed that inclusion of GSM resulted in an increase of IL-17 in the sham rats but levels were decreased in the OVX rats.

### 3.4. Analysis of Adipokines in Systemic Circulation and Fat Tissue

Adiponectin was the most abundant adipokine in the blood circulation (Figure 3). In sham rats, the lowest concentration (6.37 ± 0.59 μg/mL) was found in the ND group, and the highest concentration (7.64 ± 0.38 μg/mL) in the HFHS + GSM group. OVX significantly increased adiponectin production in all diet groups (>9 μg/mL). GSM also increased adiponectin in OVX rats fed the ND but not the HFHS diet.

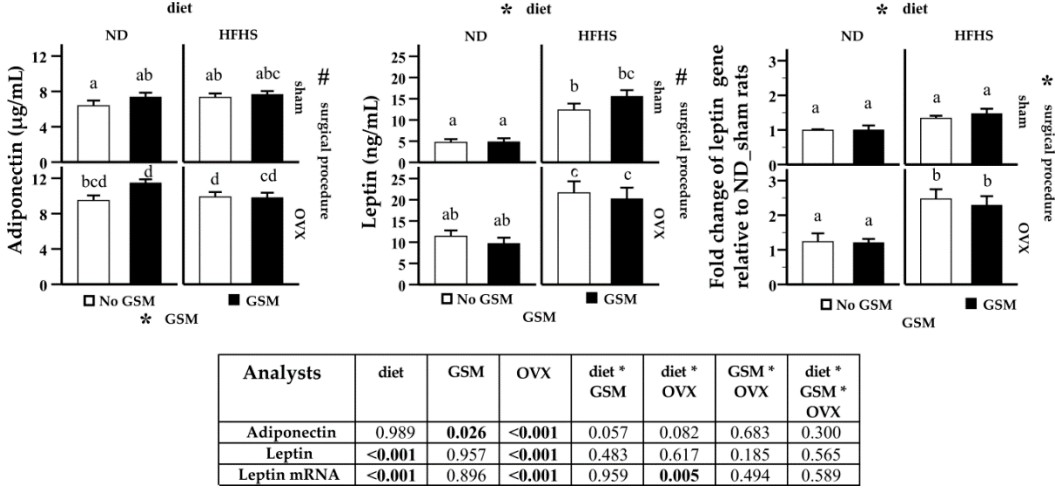

**Figure 3.** Analysis of plasma adipokines and leptin mRNA expression. Plasma adiponectin and leptin concentrations were measured using ELISA method (*n* = 11–12 per group). The quantification of Leptin mRNA expression was performed in perigonadal fat using qRT-PCR method (*n* = 7). Bar graphs and error bars show means and SE respectively. Two-way ANOVA was used to evaluate the influence of effects; diet, surgical procedure and GSM. If the effect was statistically significant then, the asterisk * or hash tag # was indicated. Having similar asterisk on the both effects means the interaction effect detected. One-way ANOVA was used to identify significant differences across all groups. In case data reached the criteria, a multiple comparison test is performed using Turkey method, and then letters are indicated on the bar. Groups showing different letters are significant differences. The table shows *p* values calculated from two-way ANOVA analysis. *p* < 0.05 was used to determine statistical significance in all methods.

The effect of HFHS diet on plasma leptin was more distinct than adiponectin. Rats fed HFHS had increased levels of leptin at more than two-fold compared to their ND counterparts, and OVX also increased leptin production similar to its effect on adiponectin. GSM however had no significant effect on leptin. To confirm the source of leptin production, the perigonadal fat pad, which attaches along the female reproductive tract, was harvested and leptin mRNA from the adipocytes was quantitatively measured. The result showed a corresponding pattern to the plasma leptin levels (Figure 3). Feeding HFHS in sham rats resulted in a measurable increase in leptin mRNA compared to the sham rats fed ND, and significantly increased leptin mRNA to more than double those of the OVX rats compared to ND OVX. The interaction of diet and OVX had an additive effect on leptin gene expression and subsequent leptin production.

## 3.5. Analysis of Cartilage Degradation and Glucose Metabolism

A significant alteration in glucose metabolism was detected by increased plasma HbA1c in OVX rats (Figure 4). OVX, diet and GSM did not independently or additively influence the cartilage degradation marker, CTX-II.

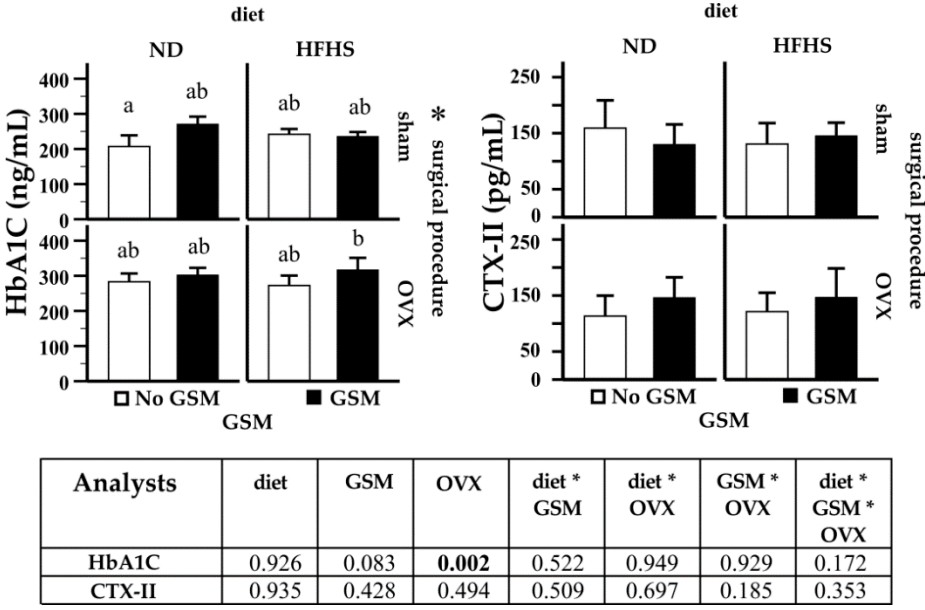

| Analysts | diet | GSM | OVX | diet * GSM | diet * OVX | GSM * OVX | diet * GSM * OVX |
|---|---|---|---|---|---|---|---|
| HbA1C | 0.926 | 0.083 | **0.002** | 0.522 | 0.949 | 0.929 | 0.172 |
| CTX-II | 0.935 | 0.428 | 0.494 | 0.509 | 0.697 | 0.185 | 0.353 |

**Figure 4.** Analysis of cartilage and glucose metabolism in systemic circulation. Plasma analysis was measured in the rats at 48 weeks of age. Bar graphs and error bars show means and SE respectively (*n* = 11–12, per group). Two-way ANOVA was used to evaluate the influence of effects: diet, surgical procedure, and GSM. If the effect was statistically significant, the symbol * is indicated. One-way ANOVA was used to identify significant differences across all groups. In case data reached the criteria, a multiple comparison test is performed using Turkey method, and then letters are indicated on the bar. Groups showing different letters are significant differences. The table shows *p* values calculated from two-way ANOVA analysis. $p < 0.05$ was used to determine statistical significance in all methods.

### 3.6. Pathological Changes in Knee Joints

Total Mankin score of knee joints examined histologically was compared in rats of different ages fed a similar diet (ND) (Figure 5). Younger rats (26 weeks old) showed lower scores than the aged rats (48 weeks old), demonstrating that age alone in the absence of HFHS diet or OVX resulted in significant knee joint damage.

| Age | Knee joint scoring | | | |
|---|---|---|---|---|
| | Structure | Cellularity | Matrix | Tide mark |
| **Young rats** | 0.53±0.31 | 0.30±0.13 | 0.00±0.00 | 0.00±0.00 |
| **Aged rats** | 0.80±0.29 | 1.40±0.16 | 1.20±0.36 | 1.00±0.26 |

**Mankin score**

**Figure 5.** Comparison of knee joint scores between young rats and aged rats. Rats on normal control diet at age of 26 weeks (young rats, *n* = 12) and 48 weeks (aged rats, *n* = 10) were sacrificed and knee joints were processed to pathological slide preparation stained with Safranin-O. The criteria of Mankin score were used to evaluate articular cartilage and then summed into mean ± SE presented in the bar chart. Mann–Whitney U test was used to analyzed statistical differences between young rats and aged rats as which asterisk *** shows significance at $p < 0.001$.

Figure 6 demonstrates the pathological findings of knee joints in the 48-week-old rats from each of the test groups, with images selected to visually represent the various criteria used to inform the Mankin score. Severity of cartilage erosion was graded from 0 to 14 with the higher scores corresponding to more severe deterioration. As can be seen, all rat groups had only modest severity (4–5) except the OVX rats fed HFHS; this group expressed a Mankin score of approximately 7, which is considered being moderate severe disease. Rats on the ND developed only modest OA with no significant differences due to other interventions. Rats on the HFHS diet that did not undergo OVX likewise had only moderate scores, although the score was decreased slightly but not significantly with GSM addition. Thus, neither OVX nor HFHS diet alone induced additional joint damage, but the combination resulted in a synergistic effect causing moderate damage. Of greatest interest was the observation that inclusion of GSM in the OVX rats fed HFHS resulted in complete mitigation of the induced pathological lesions (Figure 6).

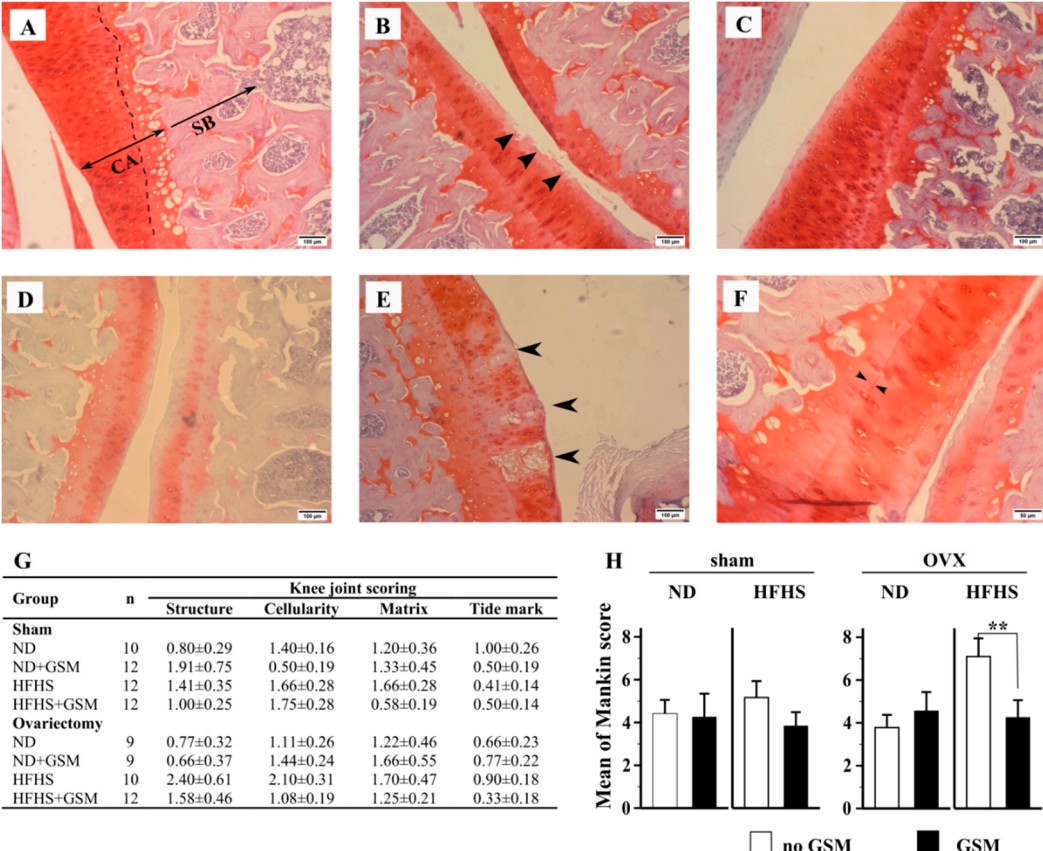

**Figure 6.** Pathological changes in rat's knee joints at 48 weeks of age. Knee joints from rats at the end of the study were dissected, formalin-fixed, and decalcified prior to processing tissue slides. Slides were stained with Safranin-O showing in panels A–F. Panel (**A**) shows normal features of rat tibia bone as which articular cartilage is stained red (CA) and subchondral and trabecular bone are pink (SB). The cartilage can be divided by tide mark (dotted line) into two zones; non-calcified cartilage (from the surface area to tide mark junction) and calcified cartilage (under tidemark to subchondral bone plates). The abnormality of articular cartilage can be marked as surface irregularity (black arrows in panel (**B**)), cloning and clusters of chondrocytes (**C**), pale staining of cartilage matrix (**D**), degeneration and necrosis of chondrocytes (black arrow in panel (**E**)), and duplicated tide mark (**F**). Mankin score criteria were used to grade the severity of cartilage lesion in each group and mean ±SE is showed in the table (**G**). Finally, all criteria were summed and presented as a mean ±SE of Mankin score in the bar chart (**H**). Mann–Whitney U test was used to analyzed statistical differences between groups and asterisk ** shows significance at $p < 0.01$.

## 4. Discussion

The increased prevalence of OA over the past decade is attributed to the expansion of both an aging society and the consumption of a "western-type" high energy diet leading to increased obesity [30]. In addition, menopause is considered one of the most significant causes of OA in women [3]. This study's experimental design was intended to assess which of these factors (aging, diet, and menopause) independently or synergistically cause OA and how they determine the disorder's signature pathological pattern. This study successfully established the multiple phenotype of OA in a female rat model by using the mixed factors of aging, obesity, and ovariectomy. Interestingly, GSM consumption directly alleviated cartilage destruction without modulation of the predisposing factors including obesity, hyperglycemia, and bone loss due to estrogen deficiency. We speculate that the biological activity of GSM can be mainly attributed to omega-3 fatty acids, docosahexaenoic acid (DHA), and eicosapentaenoic acid (EPA), which are abundant in GSM powder [31] and well known to have anti-inflammatory properties [32–36]. However, GSM also contains other nutrient compositions such as a serious of unusual lipids (non-methylene interrupted FAs and plasmalogens), proteins, carbohydrates, and insoluble fiber, which are not well elucidated in their biological activity [37,38], but these may contribute partially to some effects in this study. Therefore, the flash-dried GSM used in the current study may produce an anti-arthritis effect as effectively as GSM oil has been shown to do in previous studies [22,24] It was observed in the current study that in the early cohort rats, whole GSM powder significantly reduced CTX-II [31], an early prognostic biomarker for OA. Similarly, in the late cohort rats, whole GSM powder significantly reduced OA in rats with a high Mankin score, which was the only definitive diagnostic marker for OA.

Diet-induced obesity models have been used in rats for many decades. Feeding a diet with high proportions of fat and sugar accelerates bodyweight gain. The main bodyweight gain is attributed to an increase in body fat, the characteristic of obesity. Interestingly, inclusion of GSM in the diet of sham rats resulted in increased bodyweight but it was opposite in the OVX rats. It is possible that some components in GSM might be able to facilitate the digestion or absorption of macronutrients in the gastrointestinal tract resulting in an increase of nutrient bioavailability [39]. However, the mechanism of GSM on weight reduction in the OVX rats due to less fat mass gain cannot be explained by the current available information.

Obese rats have a substantially increased proportion of body fat, mainly in the abdominal cavity (visceral fat) and under the skin (subcutaneous fat) [40]. Visceral fat is composed of white adipocytes which are responsible to adipokine production [41]. Leptin, one of the adipokines highly correlated with body fat mass, is known to be a critical mediator that initiates low-grade chronic inflammation by activating Toll-like receptors on macrophages [42]. Subsequently, activated macrophages release various types of proinflammatory cytokines into the circulatory system, manifesting inflammatory responses [43]. In this study, plasma leptin was dramatically elevated in the HFHS fed rats proportional to bodyweight gain.

Estrogen deficiency had an explicit influence on increased bodyweight and plasma leptin [44]. To confirm the source of leptin production, the perigonadal fat pad, one portion of visceral fat, was assessed for leptin mRNA expression. A similar incremental pattern between plasma leptin and gene expression was revealed in all groups, indicating that perigonadal fat is one of the main sources of circulating leptin production in obesity which corroborates with previous rat studies [45,46]. Plasma adiponectin levels in ND sham rats were high (~7 μg/mL) and similar to those found in humans under matching physiological conditions (5–10 μg/mL) [47]. Both increased and decreased adiponectin levels correlating with obesity have been reported. Moreover, the controversy of adiponectin's effect on systemic chronic inflammation and joint inflammation is still a matter of debate [48]. Our study revealed that adiponectin was less sensitive in responding to the increased bodyweight or body fat, but ovariectomy significantly elevated adiponectin. Increased adiponectin can induce NOS2, IL-6, MMP-3, MMP-9, and MCP-1 expression in chondrocytes [49], influencing the pathophysiology of osteoarthritis.

Theoretically, low-grade inflammation in obesity is characterized by a systemic influx of adipokines (leptin, adiponectin, and resistin) and an increase of proinflammatory cytokines (IL-1β, IL-6, TNF-α, and MCP-1) in small quantities, and thus linked to many chronic inflammation diseases [50,51]. In one published study, rats fed high fat and high carbohydrate diets were shown to have significantly increased serum IFN-γ and IL-1β, and decreased serum IL-10, concurrent with synovitis and cartilage destruction [17]. Collins et al.'s [16] study, on the other hand, showed that serum IFN-γ, Il-10, IL-12, IL-13, IL-17a, IL-18, IL-1β, Il-2, IL-6TNF-α, and MCP-1 were not significantly different between diet-induced obese rats and normal-diet fed rats; however, most of the proinflammatory cytokines were elevated in synovial fluid and related to joint pathological lesions. It is possible that low-grade inflammation might be obscured at the systemic level but is definitively altering immune responses at local tissue or cellular function levels [52]. The evidence showed that activation of macrophage functions at local tissue such as synovial membrane is related to OA or rheumatoid arthritis [53]. Our results concurred with this previous work.

We did not see modulation of proinflammatory cytokines in plasma in response to HFHS, and only increased plasma MCP-1 was detected in response to ovariectomy. MCP-1, a chemokine produced by adipocytes, macrophages, and endothelial cells, can regulate macrophage polarization and infiltration, insulin sensitivity, and lipid metabolism [54]. Therefore, it causes local inflammation such as arteriosclerosis and coronary heart disease [55] or probably osteoarthritis. Interestingly, plasma MCP-1 theoretically is associated with adiposity [56], but our data revealed that MCP-1 was modified by the effect of ovariectomy, not obesity. The male rats in Collin's study [16] gained even more weight than our female rats, but no change of plasma MCP-1 levels was detected. This evidence indicates that MCP-1 is distinctly different in females and might be a crucial modulator of MetOA in post-menopausal women; if so, then it could be used as a biomarker to diagnose or monitor disease progression in this population. It may also be of interest to determine whether drugs that target MCP-1 are more potent against this specific OA phenotype.

Further, the change in plasma IL-17 influenced by the interaction effect between GSM and OVX was irrelevant to other results. IL-17 is recognized as a T cell activator of neutrophil mobilization and activation, mediating innate immunity and being involved with the pathogenesis of inflammatory diseases such as psoriasis and rheumatoid arthritis [57]. Therefore, IL-17 would have been expected to increase in the OVX rat fed HFHS, which showed pathological joints. However, a previous study revealed no relation of IL-17 to HFHS diets [16]. Despite a lack of explanation to this occurrence, we speculate that this change may be related to an alteration of gut microbiota due to GSM and OVX, or possibly an as-yet unexplored interaction between GSM, diet, and estrogen levels.

Menopause is a critical factor of osteoarthritis in women, changing the low prevalence of female at age under 50 years old to a prevalence greater than males after the postmenopausal period [3]. The result at the end of this study showed that average BMD in OVX rats was 10% lower than sham rats in all bone sites. Ovariectomy resulting in estrogen deficiency caused the significant reduction of BMD in both lumbar spine and femurs. Lack of estrogen changes the course of bone metabolism/mineralization, inducing excessive osteoclast activity and reducing calcium deposition in bone, resulting in weakening of the trabecular bone that forms the porous bone network beneath calcified cartilage to support load-bearing by the joints. Changing the microscopic structure in this area can cause the development of bone cysts, osteophyte formation, flattening and deforming of the subchondral articular contour and all these features are recognized as osteoarthritis [58].

Despite the fact that the rat model does not completely mimic human OA because of differences in growth plate activity during adulthood, patterns of cartilage loading, and spontaneous intrinsic healing of cartilage lesions [59], rats in addition to mice and rabbits [60] are the main animal models used in OA research. Their main advantages are cost efficiency and manageability, and the rat model provides crucial evidence for further study in other larger animals. This species is not considered as a spontaneous OA model like guinea pigs, STR/1N mice, or C57/BL mice, but it is commonly used in a chemical induction model using intraarticular injection of adjuvant, collagen, iodoacetate, or

papain [60,61]. Just in the last decade, high-energy-diet-induced OA models have been developed, first in mice [62], then rabbits [63], and most recently rats [15]. As a new animal model, evidence is still inconsistent. Our study revealed interesting and novel information about this model. First, young female rats (26 weeks of age) did not spontaneously evolve explicit pathological lesions in knee joints; however, the older rats (48 weeks of age) revealed some pathological changes by the Mankin score. Even though the lesions in the cartilage layers appeared to be only a modest feature of OA, the changes were significant after time; a similar finding has been reported [64]. This means that age also contributes to OA development at subclinical levels in the rat models. At 48 weeks of age, all rats showed at least mild features of cartilage attrition. As low grade inflammation due to HFHS diet and bone loss from estrogen deficiency progressed, cartilage destruction became more pronounced and distinct only in one particular group. This cumulative evidence can explain the multiple phenotypes of osteoarthritis observed in this rat model.

Interestingly, plasma concentrations of CTX-II, the cartilage degradation marker, did not correlate with the pathological changes in knee joints. In a matching study, we did detect a significant increase of CTX-II in younger rats correlating with obesity and the initiation of Met-OA [31]. A similar study showed that serum CTX-II dramatically increased in OVX rats within 3 weeks of surgery, then declined to the baseline level six weeks later OVX [65]. Likewise, Høegh-Andersen et al. [66] monitored serum CTX-II in 5-month-old ovariectomized rats and found that CTX-II transiently increased by approximately 50% within 2 weeks after surgery and then declined afterward to 50% below baseline at week 8. CTX-II in the sham rats also declined as the same level of the OVX rats. This indicates that the sensitivity of serum CTX-II as a cartilage marker is of less value in rats older than 6 months. This ambiguity is likely due to the limitation of the model, as mature rats retain growth plate function. To overcome this drawback, CTX-II levels should be monitored immediately after the second week of surgery. Another limitation of this study is due to the lack of revelation in the precise pathological mechanisms driving the development of the disease and the anti-osteoarthritis effect of GSM. This may be an interesting gap for other studies in the future. However, no adverse effects of GSM were noted, suggesting that the findings in the current pre-clinical study warrant clinical trials with whole GSM powder.

In conclusion, our data support the hypotheses that diet and OVX independently increase weight gain via body fat gain. Ageing, HFHS diet and OVX-induced estrogen deficiency together cause low-grade-chronic inflammation via leptin, adiponectin and MCP-1 upregulation. OVX-induced bone resorption can further accelerate joint instability resulting in the cartilage destruction observed in OA. Finally, GSM independently has a potential role in OA prevention.

**Author Contributions:** Conceptualization, F.M.W., M.C.K., M.R.M., and H.T.; methodology, F.M.W., M.C.K., and P.S.; formal analysis, P.S.; investigation, P.S., M.R.M., and F.M.W.; resources, F.M.W. and M.C.K.; data curation, P.S.; writing—original draft preparation, P.S.; writing—review and editing, F.M.W., M.C.K., M.R.M., H.T. and P.S.; visualization, P.S.; supervision, F.M.W. and M.C.K.; project administration, F.M.W.; funding acquisition, M.R.M., H.T., and M.C.K. All authors have read and agree to the published version of the manuscript.

**Funding:** This study was funded by the National Science Challenge, High value nutrition (HVN) "Musseling up: high-value Greenshell™ mussel foods" (UOAX1421)." MU College of Health PhD scholarship, as collaboration between Massey University, Cawthron Institute, and Sanford Ltd.

**Acknowledgments:** The authors thank Anne Broomfield (Massey University) for DXA scanning, Gabrielle Plimmer and De Shampa (Massey University) for supporting in necropsy, and Thanongsak Mamom (Mahanakorn University) for consultation in pathology.

**Conflicts of Interest:** H.T. has a conflict of interest as she is employed by Sanford Limited which is a mussel producer and exporter.

## Appendix A

**Table A1.** Body composition of the rats at 48 weeks of age.

| | Sham | | | | OVX | | | | |
|---|---|---|---|---|---|---|---|---|---|
| | **ND** | **ND + GSM** | **HFHS** | **HFHS + GSM** | **ND** | **ND + GSM** | **HFHS** | **HFHS + GSM** | ***p* Values** |
| BW 12th week | 273.73 ± 8.53 | 276.06 ± 6.20 | 280.84 ± 7.01 | 281.62 ± 7.54 | 275.64 ± 8.20 | 267.37 ± 8.96 | 276.51 ± 7.93 | 280.70 ± 6.41 | 0.911 |
| BW the end | 403.89 ± 17.30 [a] | 428.18 ± 18.05 [a] | 535.38 ± 27.16 [bc] | 578.15 ± 21.78 [bc] | 499.20 ± 18.56 [ab] | 488.27 ± 22.36 [ab] | 633.35 ± 34.01 [c] | 619.05 ± 29.34 [c] | **<0.001** |
| Change of BW (%) | 49.01 ± 4.74 [a] | 54.55 ± 3.97 [ab] | 89.68 ± 6.97 [cd] | 105.51 ± 6.86 [cde] | 80.99 ± 3.34 [bc] | 83.41 ± 7.66 [bc] | 129.03 ± 10.51 [e] | 118.15 ± 7.44 [de] | **<0.001** |
| Change of %body fat (%) | 185.55 ± 25.75 [a] | 221.94 ± 35.14 [ab] | 344.14 ± 33.06 [bc] | 377.30 ± 41.02 [c] | 328.88 ± 43.10 [abc] | 303.18 ± 18.48 [abc] | 357.20 ± 27.90 [bc] | 338.88 ± 37.36 [bc] | **0.001** |
| **Statistical analysis by two-way ANOVA with three effects** | | | | | | | | | |
| | **diet** | **GSM** | **OVX** | **diet * GSM** | **diet * OVX** | | **GSM * OVX** | **diet * GSM * OVX** | |
| BW 12th week | 0.217 | 0.964 | 0.579 | 0.615 | 0.944 | | 0.740 | 0.518 | |
| BW the end | **<0.001** | 0.538 | **<0.001** | 0.824 | 0.807 | | 0.176 | 0.748 | |
| Change of BW (%) | **<0.001** | 0.498 | **<0.001** | 0.874 | 0.642 | | 0.119 | 0.217 | |
| Change of %body fat (%) | **<0.001** | 0.792 | **0.042** | 0.966 | **0.011** | | 0.243 | 0.913 | |

Body composition of the rats was reported as mean ± SE (*n* = 11–12). Rat bodyweight was measured on scales at the beginning of the study and at the end and percent change of BW was calculated accordingly. Similarly, % body fat at the start and end from DXA scanning was reported as percent change. The statistically significant differences across all groups were analyzed using one-way ANOVA and *p* values were reported in the last column. In case there was a significant difference, multiple comparison tests, using Tukey method, was applied and the differences of superscript letters indicated the statistical significance. Two-way ANOVA was used to analyze the influence of diet, GSM and ovariectomy (OVX) on the body composition. These three effects were treated as fixed effects in general linear models. The significance of single or interaction effect is presented in bold letters. The single effects and the interaction effects are shown on the left and the right columns respectively. The *p* values were summarized in the table below. Statistical differences is applied when *p* < 0.05.

**Table A2.** Analysis of inflammatory markers in systemic circulation.

| | Sham | | | | OVX | | | | |
|---|---|---|---|---|---|---|---|---|---|
| | **ND** | **ND + GSM** | **HFHS** | **HFHS + GSM** | **ND** | **ND + GSM** | **HFHS** | **HFHS + GSM** | ***p* Values** |
| IL-1β (pg/mL) | 8.68 ± 3.63 | 7.03 ± 3.91 | 5.93 ± 1.77 | 4.48 ± 1.44 | 4.23 ± 1.61 | 7.50 ± 2.00 | 6.57 ± 1.68 | 3.73 ± 0.88 | 0.800 |
| IL-6 in (pg/mL) | 32.96 ± 4.53 | 26.79 ± 5.04 | 28.91 ± 5.35 | 27.69 ± 4.60 | 36.96 ± 9.36 | 32.44 ± 3.87 | 27.88 ± 4.27 | 28.75 ± 4.05 | 0.900 |
| IL-10 in(pg/mL) | 0.28 ± 0.29 | 1.27 ± 1.04 | 0.34 ± 0.24 | 0.07 ± 0.05 | 0.44 ± 0.32 | 0.46 ± 0.22 | 0.87 ± 0.58 | 0.17 ± 0.17 | 0.470 |
| TNF-α(pg/mL) | 1.08 ± 0.18 | 1.88 ± 0.28 | 1.78 ± 0.28 | 2.09 ± 0.18 | 1.53 ± 0.31 | 2.04 ± 0.84 | 1.89 ± 0.17 | 1.88 ± 0.56 | 0.750 |
| MCP-1(pg/mL) | 340.65 ± 49.76 [ab] | 387.69 ± 53.60 [ab] | 320.93 ± 55.72 [a] | 376.20 ± 71.69 [ab] | 483.26 ± 81.78 [ab] | 559.44 ± 69.88 [ab] | 574.41 ± 122.81 [ab] | 732.36 ± 163.99 [b] | **0.020** |
| PGE2 (pg/mL) | 1254.75 ± 187.65 | 1314.51 ± 234.97 | 1263.81 ± 212.19 | 1155.18 ± 237.85 | 1446.57 ± 221.96 | 1017.02 ± 123.24 | 1323.19 ± 313.98 | 1683.71 ± 321.77 | 0.660 |
| IFN-γ (pg/mL) | 241.55 ± 35.38 | 279.23 ± 40.09 | 222.36 ± 27.98 | 241.99 ± 35.16 | 263.20 ± 30.81 | 226.20 ± 33.19 | 217.63 ± 33.88 | 249.71 ± 26.38 | 0.695 |
| CXCL1 (pg/mL) | 23.65 ± 6.42 | 19.74 ± 5.36 | 16.83 ± 3.54 | 19.31 ± 4.88 | 32.09 ± 7.27 | 17.44 ± 4.71 | 19.71 ± 6.31 | 18.55 ± 4.82 | 0.283 |
| IL-18 (pg/mL) | 136.10 ± 48.45 | 159.41 ± 55.21 | 163.36 ± 76.23 | 112.23 ± 29.89 | 111.33 ± 13.87 | 119.65 ± 41.65 | 99.90 ± 16.12 | 130.33 ± 27.99 | 0.974 |
| IL-12 (pg/mL) | 40.30 ± 13.65 | 48.85 ± 11.20 | 34.60 ± 4.77 | 48.33 ± 5.69 | 51.25 ± 5.78 | 35.09 ± 0.76 | 65.66 ± 3.36 | 53.98 ± 5.35 | 0.804 |
| IL-17 (pg/mL) | 43.72 ± 8.31 | 50.58 ± 10.12 | 33.37 ± 6.78 | 43.40 ± 7.56 | 53.26 ± 4.60 | 33.08 ± 5.47 | 41.89 ± 10.90 | 34.59 ± 4.41 | 0.428 |
| IL-33 (pg/mL) | 612.27 ± 68.81 | 598.94 ± 95.12 | 622.62 ± 111.93 | 516.23 ± 60.95 | 650.36 ± 42.59 | 577.06 ± 80.56 | 584.99 ± 46.44 | 617.76 ± 58.09 | 0.850 |

**Table A2.** *Cont.*

| | Statistical Analysis by Two-Way ANOVA with Three Effects | | | | | | |
|---|---|---|---|---|---|---|---|
| | diet | GSM | OVX | diet * GSM | diet * OVX | GSM * OVX | diet * GSM * OVX |
| IL-1β (pg/mL) | 0.316 | 0.689 | 0.540 | 0.379 | 0.563 | 0.597 | 0.347 |
| IL-6 in (pg/mL) | 0.297 | 0.468 | 0.525 | 0.497 | 0.528 | 0.805 | 0.977 |
| IL-10 in (pg/mL) | 0.375 | 0.970 | 0.995 | 0.085 | 0.256 | 0.220 | 0.646 |
| TNF-α (pg/mL) | 0.338 | 0.171 | 0.661 | 0.390 | 0.542 | 0.607 | 0.982 |
| MCP-1 (pg/mL) | 0.367 | 0.193 | **0.001** | 0.727 | 0.253 | 0.609 | 0.775 |
| PGE2 (pg/mL) | 0.565 | 0.863 | 0.480 | 0.363 | 0.311 | 0.976 | 0.163 |
| IFN-γ (pg/mL) | 0.400 | 0.573 | 0.760 | 0.583 | 0.712 | 0.504 | 0.350 |
| CXCL1 (pg/mL) | 0.233 | 0.267 | 0.593 | 0.200 | 0.794 | 0.354 | 0.646 |
| IL-18 (pg/mL) | 0.858 | 0.925 | 0.343 | 0.651 | 0.868 | 0.565 | 0.405 |
| IL-12 (pg/mL) | 0.436 | 0.871 | 0.333 | 0.778 | 0.262 | 0.161 | 0.984 |
| IL-17 (pg/mL) | 0.189 | 0.609 | 0.690 | 0.438 | 0.711 | **0.036** | 0.639 |
| IL-33 (pg/mL) | 0.659 | 0.468 | 0.719 | 0.951 | 0.828 | 0.719 | 0.370 |

Plasma was recovered from the rat's blood at termination (48 weeks of age). Technically, ELISA methods were employed to quantify the concentration levels of all inflammatory cytokines. Data were present as mean ± SE (*n* = 11–12). The statistically significant differences across all groups were analyzed using one-way ANOVA and p values were reported in the last column. In case there was a significant difference, multiple comparison tests, using Tukey method, was applied and the differences of superscript letters indicated the statistical significance. Two-way ANOVA was used to analyze the influence of diet, GSM and ovariectomy (OVX) on the cytokines. These three effects were treated as fixed effects in general linear models. The significance of single or interaction effect is presented in bold letters. The single effects and the interaction effects are shown on the left and the right columns respectively. The *p* values were summarized in the table below. Statistical differences were applied at *p* < 0.05.

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
