# Peer review of "Effects of Greenshell Mussel (Perna canaliculus) Intake on Pathological Markers of Multiple Phenotypes of Osteoarthritis in Rats"

_applsci, doi:10.3390/app10176131_

Round 1

Reviewer 1 Report

  1. This study’s experimental design was intended to assess which of these factors (aging, diet, menopause) independently or synergistically cause ostearthritis and how they determine the disorder’s signature pathological pattern. This study successfully established the multiple phenotype of ostearthritis in a female rat model by using the mixed factors of aging, obesity and ovariectomy.
  2. On line 28, you mention the abbreviation "OA" for the first time but do not describe it. Please add that abbreviation on line 17, just after "ostearthritis".
  3. On line 32, idem. Add abbreviation "OA" for the first time after the term "ostearthritis".
  4. Why did you use 10% GSM powder and no more? Have you checked with different dose-response ratios? Or on the other hand, have you tried using a high fat diet where your 30% of the fat composition of the diet contains only GSM? In other words, substitute soybean oil and lard for GSM (100%).
  5. References are cited in Vancouver, but please, in the bibliography section indicate at the beginning of each reference their corresponding enumeration to facilitate the search.
  6. The differences between normal diet (ND) sham and OVX controls are obvious. However, why was the ND control (26 weeks old) not operated (sham)? To study aging as another factor, unless you had replicated a simulation of the operation (26 weeks old SHAM vs. 48 weeks old SHAM). Or at least, have also included non-operating control animals in both age ranges.
  7. Both in tables and figures, the significant differences between groups are well indicated. However, the interactions of the superscript letters are unclear. Please recommend in the supplementary material to clarify a little better with which groups implies the significant differences of each superscript letter. It would further facilitate understanding of the results.
  8. Lines 208-210: “There was also an interaction effect of GSM and OVX on IL-17. The result showed that inclusion of GSM resulted in an increase of IL-17 in the sham rats but levels were decreased in the OVX rats”. Also, you did not see modulation of pro-inflammatory cytokines in plasma in response to HFHS. What can be the reason for the elevation of plasma IL-17 levels only in Sham rats?
  9. Please do not see the dotted lines on the photomicrographs in Figure 6 as defined in line 284.
  10. Lines 274-277: “Thus, neither OVX nor HFHS diet alone 274 induced additional joint damage, but the combination resulted in a synergistic effect causing 275 moderate damage. Of greatest interest was the observation that inclusion of GSM in the OVX rats 276 fed HFHS resulted in complete mitigation of the induced pathological lesions”. The results show that GSM cartilage erosion and also protects against joint damage observed with the Mankin Score Criterion. It is one of the most relevant results.
  11. Continuing with the aforementioned, GSM consumption directly alleviated cartilage destruction without modulation of the predisposing factors including obesity, hyperglycaemia, and bone loss due to estrogen deficiency. You speculate that the biological activity of GSM can be mainly attributed to omega-3 fatty acids, docosahexaenoic acid (DHA) and eicosapentaenoic acid (EPA) which are abundant in GSM powder and well known to have anti-inflammatory properties.
  12. Lines 312-315: "Interestingly, inclusion of GSM in the diet of sham rats resulted in increased bodyweight but it was opposite in the OVX rats. It is possible that some components in GSM might be able to facilitate the digestion or absorption of macronutrients in the gastrointestinal tract resulting in an increase of nutrient bioavailability". These data are very interesting from a clinical point of view. In principle, the results, together with the higher levels of hemoglobin glycosylase (HbA1C) in HFHS + GSM in OVX rats, do not have to be negative. It may be a sign that there is no more glucose, but that you use less glucose and have an increased use of omega-3 fatty acids. Still, have you checked whether together with elevated leptin levels (plasma and mRNA) in HFHS diets of OVX rats (with / without GSM) they manifest insulin resistance? Have you done the homeostatic model assessment (HOMA-IR)?
  13. You argue in the discussion (lines 347-350) most of the proinflammatory cytokines were elevated in the synovial fluid and relate it to pathological joint lesions. With the Mankin Score, other authors relate osteoarthritis due to synovial macrophages that can regulate macrophage activation and infiltration of this barrier (Culemann et al., 2019). Do you have any indication that the changes produced in the joint by OVX (and reversed with GSM) are a consequence of a direct relationship of the proinflammatory activity of synovial macrophages that form in the immunological barrier of the synovial membrane (linning layer)? Source: Culemann et al., 2019. https://doi.org/10.1038/s41586-019-1471-1
  14. Lines 394-395: “Interestingly, plasma concentrations of CTX-II, the cartilage degradation marker, did not correlate with the pathological changes in knee joints”. ……… Lines 401-404: “CTX-II in the sham rats also declined as the same level 401 of the OVX rats. This indicates that the sensitivity of serum CTX-II as a cartilage marker is of less 402 value in rats older than 6 months. This ambiguity is likely due to the limitation of the model, as 403 mature rats retain growth plate function”.
  15. CTX-II did not correlate with the pathological changes in knee joints, possibly to the variability found in your samples, or to the fact that it stabilized until reaching its basal level since it was operated (OVX) until the end of the experimental period. In this case, measurement should be considered within the first two weeks after the operation.

Author Response

We greatly appreciate your helpful comments.  We have addressed or responded all those comments as shown below.

Reviewer1_comments and suggestions for authors

This study’s experimental design was intended to assess which of these factors (aging, diet, menopause) independently or synergistically cause osteoarthritis and how they determine the disorder’s signature pathological pattern. This study successfully established the multiple phenotype of osteoarthritis in a female rat model by using the mixed factors of aging, obesity and ovariectomy.

  1. On line 28, you mention the abbreviation "OA" for the first time but do not describe it. Please add that abbreviation on line 17, just after "osteoarthritis".

Author:  This change has been made in the manuscript.

  1. On line 32, idem. Add abbreviation "OA" for the first time after the term "osteoarthritis".

Author: This change has been made in the manuscript

  1. Why did you use 10% GSM powder and no more? Have you checked with different dose-response ratios? Or on the other hand, have you tried using a high fat diet where your 30% of the fat composition of the diet contains only GSM ? In other words, substitute soybean oil and lard for GSM (100%).

Author: Due to the experimental design introducing many conditions in the study, it was not feasible to test with several doses. Therefore, we selected the probable dose which was supposed to use in humans after applying the species conversion ratio. The purpose of using high fat diet was to induce obesity leading to osteoarthritis in rats but GSM was an intervention to prevent OA.  The diets calories are equivalent between ND and ND+GSM or HFHS and HFHS+GSM as they were predetermined in the formula.  It is not possible to substitute all fat in the diet with GSM powder as GSM powder only has 8% fat. It is also too expensive to use the GSM oil in the diet formula.  

This has been addressed in the manuscript in additional text in what are now lines 72 – 85 in the introduction.

  1. References are cited in Vancouver, but please, in the bibliography section indicate at the beginning of each reference their corresponding enumeration to facilitate the search.

Author: This change has been made in the manuscript

  1. The differences between normal diet (ND) sham and OVX controls are obvious. However, why was the ND control (26 weeks old) not operated (sham)? To study aging as another factor, unless you had replicated a simulation of the operation (26 weeks old SHAM vs. 48 weeks old SHAM). Or at least, have also included non-operating control animals in both age ranges.

Author: There is a concern about long term effects of anaesthesia and surgery-induced discomfort and in animals on the alteration of bone mineral density, therefore sham rats are an appropriate control to use for OVX rats in the long term cohorts. It is not sufficiently justified to utilise an additional set of no-surgery rats for this cohort given the main aims of the study as this would challenge the principle of animal ethics (3Rs).  The younger rats were used to compare the effect of age on joint pathology but not bone mineral density loss so there was no need to introduce pain from surgery to these animals. 

  1. Both in tables and figures, the significant differences between groups are well indicated. However, the interactions of the superscript letters are unclear. Please recommend in the supplementary material to clarify a little better with which groups implies the significant differences of each superscript letter. It would further facilitate understanding of the results.

Author: we have added the sentences in the caption of TableA1 and 2                                                 “The significance of single or interaction effect is presented in bold letters. The single effects and the interaction effects are shown on the left and the right columns respectively”

  1. Lines 208-210: “There was also an interaction effect of GSM and OVX on IL-17. The result showed that inclusion of GSM resulted in an increase of IL-17 in the sham rats but levels were decreased in the OVX rats”. Also, you did not see modulation of pro-inflammatory cytokines in plasma in response to HFHS. What can be the reason for the elevation of plasma IL-17 levels only in Sham rats?

Author: IL-17 is recognized as T cell activation to neutrophil mobilization and activation. As such, IL-17 can mediate protective innate immunity to pathogens or contribute to the pathogenesis of inflammatory diseases, such as psoriasis and rheumatoid arthritis. However, the controversial data still exists. Collin study (2015) found no differences of IL17 levels between the normal diet and HFHS diet fed rats. We have no certain evidence to support the modified IL-17 by this interaction effect yet; however we speculate that gut microbiota may contribute to this change.  We have addressed this by adding text in the discussion (now lines 387 – 395) 

  1. Please do not see the dotted lines on the photomicrographs in Figure 6 as defined in line 284.

Author: We have increased the line thickness to make it more visible.

  1. Lines 274-277: “Thus, neither OVX nor HFHS diet alone 274 induced additional joint damage, but the combination resulted in a synergistic effect causing 275 moderate damage. Of greatest interest was the observation that inclusion of GSM in the OVX rats 276 fed HFHS resulted in complete mitigation of the induced pathological lesions”. The results show that GSM cartilage erosion and also protects against joint damage observed with the Mankin Score Criterion. It is one of the most relevant results.

Author: We thank the reviewer for their positive and supportive remarks.

Continuing with the aforementioned, GSM consumption directly alleviated cartilage destruction without modulation of the predisposing factors including obesity, hyperglycaemia, and bone loss due to estrogen deficiency. You speculate that the biological activity of GSM can be mainly attributed to omega-3 fatty acids, docosahexaenoic acid (DHA) and eicosapentaenoic acid (EPA) which are abundant in GSM powder and well known to have anti-inflammatory properties.

Lines 312-315: "Interestingly, inclusion of GSM in the diet of sham rats resulted in increased bodyweight but it was opposite in the OVX rats. It is possible that some components in GSM might be able to facilitate the digestion or absorption of macronutrients in the gastrointestinal tract resulting in an increase of nutrient bioavailability". These data are very interesting from a clinical point of view. In principle, the results, together with the higher levels of hemoglobin glycosylase (HbA1C) in HFHS + GSM in OVX rats, do not have to be negative. It may be a sign that there is no more glucose, but that you use less glucose and have an increased use of omega-3 fatty acids. Still, have you checked whether together with elevated leptin levels (plasma and mRNA) in HFHS diets of OVX rats (with / without GSM) they manifest insulin resistance? Have you done the homeostatic model assessment (HOMA-IR)?

Author: It would be a very positive finding if omega3 in GSM could manifest protection against insulin resistance as your speculation. Unfortunately, we didn’t measure HOMA-IR because the parameter was not of sufficient importance to warrant the ethical implications on rat welfare. We however evaluated the blood glucose at the end of the study which showed the similar pattern to HbA1C levels so this might not be the case.

  1. You argue in the discussion (lines 347-350) most of the proinflammatory cytokines were elevated in the synovial fluid and relate it to pathological joint lesions. With the Mankin Score, other authors relate osteoarthritis due to synovial macrophages that can regulate macrophage activation and infiltration of this barrier (Culemann et al., 2019). Do you have any indication that the changes produced in the joint by OVX (and reversed with GSM) are a consequence of a direct relationship of the proinflammatory activity of synovial macrophages that form in the immunological barrier of the synovial membrane (linning layer)? Source: Culemann et al., 2019. https://doi.org/10.1038/s41586-019-1471-1

Author: This is also an interesting area in which if we have the opportunity and funding, we would gladly investigate. However, the objective of this study was to focus on the phenomena of biomarkers so the underlying pathological mechanisms will we hope be addressed in a subsequent study. We declared that this gap was a limitation of the study in the discussion (now lines 437-432).       

  1. Lines 394-395: “Interestingly, plasma concentrations of CTX-II, the cartilage degradation marker, did not correlate with the pathological changes in knee joints”. ……… Lines 401-404: “CTX-II in the sham rats also declined as the same level 401 of the OVX rats. This indicates that the sensitivity of serum CTX-II as a cartilage marker is of less 402 value in rats older than 6 months. This ambiguity is likely due to the limitation of the model, as 403 mature rats retain growth plate function”.

CTX-II did not correlate with the pathological changes in knee joints, possibly to the variability found in your samples, or to the fact that it stabilized until reaching its basal level since it was operated (OVX) until the end of the experimental period. In this case, measurement should be considered within the first two weeks after the operation.

Author: Thank you for this excellent suggestion. Yes, it is possible that the timing that we picked up for CTX-II was not appropriate. We declared that this gap was a limitation of the study in the discussion (now lines 437-432).   

Reviewer 2 Report

Authors explored the anti-osteoarthritic effect of GSM in animal models. Experimental approach was clear and performed logically, however, need to clarify the originality and novelty of this study. greenshell mussel is a popular source of medicinal plant and similar study on anti-osteoarthritic was already reported. Authors should clarify the difference, novelty and scientific advancement compared to the previous studies for the publication in Applied Sciences.
1. The author please indicate in the table to two decimal places. (figure 5)
2. In figure 6, as the method of quantitative evaluation of the image is described, it is necessary to add a representative photo for each group.
3. In 374-383 lines, describe several animal models of OA and many examples of experiments. why wasn't the model derived from OA used in this study? Although aging, diet, and OVX are all factors that make osteoarthritis worse, isn't it more reasonable to induce a uniform level of osteoarthritis and observe the effects at all?
4. The author can find a lot of papers that have experimented with OA animal models using GSM. It is necessary to describe preceding studies on the effects of GSM on OA models and to compare and consider the results of this study.
5. In Fig 6H, There was no statistically significant difference in ND, but rather, the control group tended to have a higher Mankin score than the GSM group. This needs to be explained.
6. GSM seems to be effective only when OVX, HFHS, and old age are all reflected, but it is interpreted as meaning that it is not effective for those who do not. This needs to be discussed.
7. What are the limitations of GSM for the treatment of osteoarthritis? and describe the direction to be presented.
8. Please check the typo: Turkey  Tukey

Author Response

We greatly appreciate your helpful comments.  We have addressed or responded all those comments as shown below.

Reviewer 2_Comments and Suggestions for Authors

Authors explored the anti-osteoarthritic effect of GSM in animal models. Experimental approach was clear and performed logically, however, need to clarify the originality and novelty of this study. greenshell mussel is a popular source of medicinal plant and similar study on anti-osteoarthritic was already reported. Authors should clarify the difference, novelty and scientific advancement compared to the previous studies for the publication in Applied Sciences.

  1. The author please indicate in the table to two decimal places. (figure 5)

Author: This change has been made in the manuscript

  1. In figure 6, as the method of quantitative evaluation of the image is described, it is necessary to add a representative photo for each group.

Author: This was one of our options to present the pathological findings. However, we determined that the Mankin score was a more quantitative and useful way of presenting the data within each group, with the images used to illustrate the Mankin criteria. Therefore, we decided to put pictures in the way that we can give more explanation to readers; this approach is also common in the literature in this area.  We have added an explanatory sentence to this effect to the text, now lines 283-285.

  1. In 374-383 lines, describe several animal models of OA and many examples of experiments. why wasn't the model derived from OA used in this study? Although aging, diet, and OVX are all factors that make osteoarthritis worse, isn't it more reasonable to induce a uniform level of osteoarthritis and observe the effects at all?

Author: There are many phenotypes of OA which may originate from different causes. The works in the past using anti-inflammatory models or post-traumatic OA models did not represent the metabolic OA. In addition, the aim of this study was not due to only the testing of GSM properties but is also the study in complex relationship of those factors driving MetOA, which manifests with a wide range of severity in humans. This is partially explained in the introduction (lines 51 – 66) but has been further explained with additional text (now lines 81 – 87).

  1. The author can find a lot of papers that have experimented with OA animal models using GSM. It is necessary to describe preceding studies on the effects of GSM on OA models and to compare and consider the results of this study.

Author: We have added text to explain this in the discussion (“Therefore, the flash-dried GSM used in current study may produce an anti-arthritis effect similar to GSM oil in the previous studies [22,24]”) in lines 326 – 331 as well as some explanatory text in the introduction (now lines 74 – 80).

  1. In Fig 6H, There was no statistically significant difference in ND, but rather, the control group tended to have a higher Mankin score than the GSM group. This needs to be explained.

Author: We have added explanatory text in lines 288 – 291.

  1. GSM seems to be effective only when OVX, HFHS, and old age are all reflected, but it is interpreted as meaning that it is not effective for those who do not. This needs to be discussed.

Author: Similar to the above, the traditional arthritis models in pharmacological testing may not fit to the goal of this study. The rats in this study developed OA in concert with the aforementioned factors, therefore the clinical OA lesions were seen in only one rat group which was mitigated by GSM. Other groups might develop very mild lesions which mean a normal or subclinical OA therefore it is possible to see no changes in those groups due to GSM.   We have added explanatory text in the discussion in lines 326 – 331.

  1. What are the limitations of GSM for the treatment of osteoarthritis? and describe the direction to be presented.

Author: We did not detect any adverse effects in the rats consumed GSM and we agree that the manuscript needed a suggestion for future work. We have added explanatory text in the discussion in lines 437 – 441.

  1. Please check the typo: Turkey  Tukey

Author We thank you for catching this error! This change has been made in the manuscript

Round 2

Reviewer 2 Report

The requested modifications were well reflected.